# Self-Care and Sense of Coherence: A Salutogenic Model for Health and Care in Nursing Education

**DOI:** 10.3390/ijerph19159482

**Published:** 2022-08-02

**Authors:** Natura Colomer-Pérez, Joan J. Paredes-Carbonell, Carmen Sarabia-Cobo, Sergio A. Useche, Vicente Gea-Caballero

**Affiliations:** 1Department of Nursing, Faculty of Nursing and Chiropody, Frailty Research Organized Group (FROG), Universidad de Valencia, 46003 Valencia, Spain; natura.colomer@uv.es; 2Public Health Department, Conselleria Sanitat Universal i Salut Pública, GVA, Foundation for the Promotion of Health and Biomedical Research of Valencia Region (FISABIO), 46010 Valencia, Spain; joanparedes2@gmail.com; 3Faculty of Nursing, Nursing Research Group (IDIVAL), Universidad de Cantabria, 39008 Cantabria, Spain; 4Faculty of Psychology, Universidad de Valencia, 46003 Valencia, Spain; sergio.useche@uv.es; 5Faculty of Health Sciences, International University of Valencia, 46002 Valencia, Spain; vicentegeacaballero@gmail.com; 6PBM Group, Research Institute Idi-Paz, 28029 Madrid, Spain

**Keywords:** self-care, sense of coherence, salutogenic model of health, mental health, nurse, certified nursing assistant, students

## Abstract

Background: Effective advocacy on self-care and the enhancement of a sense of coherence among nurses don’t only benefit control over one’s health and personal performance, but it may have a direct impact on clinical care and on the entire healing system. In this regard, nursing curricula grounded on a salutogenic model of health (SMH) operate with strategies to engage students in self-care and contribute to improving their mental health and wellbeing. The aim of this study was to explore the relationships between self-care agency and a sense of coherence as dependent variables and the age and self-reported academic performance as independent variables in nursing assistant students. Methods: For this cross-sectional study, data were collected from a full sample of 921 Certified Nursing Assistant (CNA) Spanish students. A self-administered questionnaire, including sociodemographic variables, the ‘appraisal of self-care agency’ (ASA), and the ‘sense of coherence’ (SOC) constructs, was administered. Results: Older participants presented significantly stronger values of both constructs. Apart from a significant and positive correlation between ASAS and SOC, ANOVA analyses indicate significant differences in terms of academic performance according to different ASAS and SOC degrees. Conclusions: The findings of this study endorse the assumption that there is a consistent relationship between ASA and SOC constructs that might, indeed, have a potential effect on students’ academic performance. In practical terms, it seems relevant to try to recognise the students’ self-care agency and the sense of coherence as forceful predictive variables of mental health and wellbeing, in addition to academic success as a strength implied in the future career achievement.

## 1. Introduction

Health professionals should oversee the directing of self-care interventions for the general population [1,2]. Self-care is defined as “the ability of individuals, families and communities to promote health, prevent disease and maintain health, but also incorporates elements of mental, emotional, spiritual and professional wellbeing” [3]. Not for nothing, self-care is one of the key aims of current health programs, as indicated in the literature reviews on mental and chronic diseases [4,5,6,7], considering these ailments as the most prevalent and those with the most detrimental implications for health on a global level [8,9]. On this note, numerous health strategies for disease prevention and control are being developed, such as new models specifically directed at self-care measures and their components [10,11,12]. In parallel, there has been a rising number of studies exploring how health professionals understand concepts, such as their own self-care [13,14]. The reports on this subject are striking. Although nurses are aware of the measures necessary to maintain a healthy lifestyle, the literature indicates that they have difficulties practicing self-care for their benefit [15,16]. Furthermore, several studies from different countries have highlighted the increase in obesity, cardiovascular problems, and other chronic illnesses among nurses [17,18,19]. This phenomenon seems to be related to numerous stressors which nurses must face throughout their professional life, such as academic pressure, difficult emotional situations, work environment, and conditions with special relevance to mental health [20,21]. The study of self-care measures is, therefore, not only influenced by knowledge and lifestyle, or the characteristics of each country and culture, but also by the cognitive and psychological tools available to face situations that generate emotional imbalance [22,23]. To date, the studies on self-care reflect the need to find explanations to understand why nurses have problems in caring for their health [13,24]. They also indicate the need to consider that nurses who provide direct care will often act as role models by educating the public on health matters [25]. Thus, in recent years, the approach seems to suggest a search for applied theoretical models to analyse how to focus on self-care based on other explanatory factors, beyond the education on lifestyles strategies, in order to improve mental health and the wellbeing of nurses and future ones.

### The Salutogenic Model of Health Addressing Mental and Chronic Health Conditions

Of all the current health and healing models, the salutogenic model of health (SMH) is highlighted for its concern for and role in health management, leading to a notable increase of studies on this subject [26,27,28,29]. This model is concerned with the relationship between how people approach stressful situations (such as mental and chronic health conditions and their consequences) and the individual ability to self-manage and cope with these situations [30]. The salutogenic paradigm has marked the development of the construct called a sense of coherence (SOC), which is directly related to the ability to employ cognitive, affective, and behavioral strategies, which help to improve the ability to overcome stress [31]. The creator of the model, Antonovsky (1987), defined a sense of coherence as: “An overall orientation that expresses the degree to which one has a generalised and stable, yet dynamic, sense of confidence that (1) stimuli derived from internal and external environments in the life course are understandable, structured, predictable, and explainable; (2) resources are manageable and available with respect to the demands that the individual will encounter derived from those stimuli; and (3) those demands are challenging, meaningful, worthy of investment and commitment” [32].

Several international cross-sectional studies on the salutogenic theory have demonstrated that a strong SOC improves wellbeing and the management of complex health situations, which is particularly relevant in the case of mental health and chronic diseases [33,34,35]. In this sense, the concept of a sense of coherence has also been widely used within the nursing context and has been identified as an important factor for coping skills. It is considered a protective variable for nurses’ mental health in the face of stress and traumatic events [36]. Increasing a SOC among nurses can enhance commitment to their work in the institution and build more engaged teams [37], and it is considered one of the most important predictors of post-traumatic stress symptoms [38]. Furthermore, a sense of coherence is nowadays widely recognised in the literature as a core factor for the delivery of person-centered care [39]. In the healthcare field, the salutogenic paradigm can be developed either to design health interventions or to redirect research toward this kind of care [40]. According to this applied model, people should be accompanied in their search for the best health status. However, this process should also be based on identifying the health assets or resources related to treatment approaches, as well as for preventive approaches and health promotion per se, that maintain and improve health, mental health, and wellbeing [41]. The salutogenic model of health is, consequently, more centered on preventive aspects than pathological ones and, therefore, on having a substantial impact on the improvement of people’s health [42] because a healing healthcare system is one in which patients are supported in efforts to understand their healthcare experiences as orderly, manageable, and meaningful. Indeed, acting salutogenically implies a shift of the paradigm by advocating for a participatory approach of both individuals and communities [43]. As an example, healing-oriented practices and environments founded on the principles of salutogenesis are being introduced into the American healthcare system as the central way to sustain, support, and enrich the quality of life of care providers and users [29]. Focusing on the production of health and encouraging a movement towards health leads to comprehensive reflections. If health professionals must educate people, groups, and communities on self-care measures, it is logical to think that they must learn to take care of themselves in order to coherently convey the ability to self-manage one’s care [44,45]. Currently, multiple efforts are being made to improve the health of caring professionals [46,47]. However, there are still few studies on nursing students’ health behaviours about self-care [48,49,50] despite the fact that these are the next generation of employees in the health system [51,52]. In addition, it is interesting to analyse the process, which relates self-care with the SMH during the studying years, because a powerful SOC seemingly leads to appropriate self-care [53]. To the best of our knowledge, few studies have analysed the relationship between a SOC and self-care in health professionals and students [53,54,55]. Indeed, the few studies available suggest that it is necessary to explore this relationship further in order to enhance the current understanding and examine how to improve the self-care of health professionals and, likewise, assist patients [53,56]. This could result in the implementation of strategies more focused on providing cognitive and emotional tools for people and not only the exclusive promotion of improved lifestyles. Thus, if we find the key to harness self-care, perhaps by improving a SOC, a major impact can be made on the control of mental and chronic health conditions, for example, currently representing the most urgent focus of all health policies worldwide [57]. To reach this type of ambitious outcome, it is necessary to begin by examining the appraisal of self-care and a SOC in students based on the most immediate variables available for them—their academic performance, as this seems to have a direct relationship with subsequent professional development and with the adoption of healthy lifestyles [58,59,60]—and their age and gender [61]. Therefore, this study aimed to research the possible relationships between gender, and self-reported academic performance as independent variables, and self-care agency and a sense of coherence as dependent variables in nursing assistant students.

## 2. Materials and Methods

### 2.1. Design and Study Setting

For this cross-sectional and descriptive research, we used a full sample of CNA students voluntarily responding to the questionnaire form. This research is part of a project divided into two parts: the first one was aimed at measuring the sense of coherence among nursing assistant students (see Colomer-Pérez et al. [50]), and the second part focused on self-care agency, their relationship with the sense of coherence, and the implications on salutogenic-oriented curricula in nursing science.

### 2.2. Participants and Procedure

A total of 921 students from the entire public Health-VET schools were finally accepted to participate and fill out the questionnaire form. For the data collection time, the study participants were enrolled in the last semester of the theoretical part of the training program for a Nursing Assistant certificate degree. A CNA degree in Spain is completed in as few as 15 months through a program based on theoretical and practice contents. The participants were gathered from a total of the 23 public upper secondary schools providing health vocational education and training certifications in the Valencian community (Spain). It is noteworthy that the educational system, the curricula, and the duration of the training program are the same for all educative centers offering the CNA certification in this region. The whole study population (including all these schools) comprises a total of n = 1150 individuals. As this study was grounded on a convenience (nonprobability) sampling instead of a probabilistic design [62], there was an estimated response rate (i.e., about 80% of the potential participants) as an alternative to a minimum sample calculation used otherwise. As it covered almost the whole study population, a non-probabilistic method was used, as the research team directly contacted all the potential institutional participants, i.e., the educational centers offering this program across the region, and all of them accepted to partake in the study. There were no exclusion criteria to participate, apart from not accepting to participate in the study. The questionnaire was completely anonymous, and not participating in the study did not imply adverse consequences for the students. At the individual level, 80% of the CNA students accepted to collaborate and filled out the questionnaire form. According to the sociodemographic data gathered from the participants, the average age was 28.52 (SD = 11–42 years), 81.5% of the participants were women, 18.5% were men, 75.89% reported to be unemployed, 48.21% affirmed to have chosen this career path driven by a vocational and motivational component, and 56.46% were living in urban areas at the moment of participation in this study.

### 2.3. Measures

First, the independent variables of the study were defined as: (a) age, and (b) self-reported academic performance. The academic record in the Spanish education system is generally scored on a numerical scale of 0–10, with 10 being the highest score to reach and below 5 is considered a failed outcome. The self-reported academic performance was measured when students were asked about their academic record at the end of the last semester—when the institution had already communicated the global scores obtained—and the responses were classified among the following levels: fail (<5), pass (5–5.9), good (6–6.9), remarkable (7–7.9), outstanding (8–8.9), and with distinction (9–10). The amount of data was concretely collected at the end of the first course because the whole second course is completely developed in health institutions with a practical program, which is qualified with an “apt” qualitative record, non-affecting the global numerical academic transcript.

On the other hand, the dependent variables of the study were defined as the appraisal of self-care agency, measured through the ASA scale, ASAS [63]. The ASA scale is presented in two versions: 24-item (used for the present study) and 15-item. This questionnaire has been translated and adapted/validated to several languages in Europe, East Asia, and Latin America. Both versions of the scale have been validated in studies conducted in Brazil [64], Colombia [65], and others [66,67], and it has also been recently validated for the Spanish population [68]. The scale consists of 24 items developed on a Likert scale with five alternatives, one being “Strongly disagree,” which coincides with the lowest value of self-care agency, and the 5, “Strongly agree,” being the highest. Each individual can obtain a real score ranging from 24 to 120 points. The version used in our study obtained Cronbach’s alpha values between α = 0.70 and α= 0.82 in the different studies in which it was used. In the case of this study, we obtained Cronbach’s alpha = 0.73.

The dependent variable of the sense of coherence was assessed by means of the orientation to life questionnaire-13 item (OLQ-13, widely known as a SOC) [32]. The instrument aims to assess a global orientation of the personality that facilitates the solution of problems in an adaptive way in stressful situations to which people are subjected throughout their lives. As in the extensive questionnaire (29-item), the 13-item test also measures three essential dimensions of the SOC: (a) comprehensibility (5-item), which refers to the extent to which one perceives the stimuli that confront one as consistent, structured, and clear; (b) manageability (4-item), which is the extent to which one perceives that the resources at one’s disposal are adequate to meet life’s demands; and (c) meaningfulness (4-item), which refers to the extent to which one feels that life makes sense emotionally [32]. The answers offer a continuum of an agreement to disagreement in 7 response options represented on a Likert scale from 1 to 7 and ranging from “Never” and “Rarely” to “Very often” and “Always,” both in the sense of the positive or negative questions. A total sum was calculated, ranging from 13 to 91 points. The SOC scale has shown good internal consistency, with Cronbach’s alpha values between α= 0.70 and α= 0.92 [27,32]. It retains the same psychometric qualities as the original version of the 29 items [27,69]. In the case of this study, we obtained Cronbach’s alpha = 0.79.

### 2.4. Data Processing

Statistical analysis was carried out with the IBM SPSS statistical package, v. 22. The default confidence level was established at 95% (*p* < 0.05). Descriptive analyses were carried out for all variables, and a correlational (Pearson’s *r*) test was performed in order to assess the bivariate relationships among the dependent (the SOC and ASAS as the dependent variables.) and independent variables (age and academic performance). Additionally, a one-way analysis of variance (ANOVA) was carried out in order to compare the academic performance of individuals in accordance with their SOC and ASAS degree, using the median (50th percentile) of the distribution as the splitting criterion.

### 2.5. Ethics Statement

Ethical permissions needed were requested and obtained from the educational centers and the competent institution in the area of education governance in the region (IRB 5ED01Z/2016/406/S). Likewise, the paper questionnaires were exclusively identified by means of a generic code in order to preserve the confidentiality and anonymity of the information given by the participants. The center’s staff members and students received pertinent information on the purpose of the research and the strictly academic use of the obtained data. Responding to the questionnaire was interpreted as granting consent to participation in the research. The data were entered into a database to which only the principal investigator had access. The study was designed in accordance with the current Spanish and European data protection regulations and ethical guidelines for research involving human subjects.

## 3. Results

### 3.1. Appraisal of Self-Care Agency and and Sense of Coherence 

The descriptive analysis of the sample has already been published in Colomer-Pérez & Useche [68]). Therefore, in the present paper, we limit ourselves to carrying out: (1) the correlation tests (Pearson) in order to assess the bivariate relationships among the key study variables; and (2) an analysis of variance (ANOVA) in order to compare the academic outcomes according to the different sense of coherence (SOC) and self-care agency (ASAS) degrees.

### 3.2. Appraisal of Self-Care Agency and Academic Success: Descriptive Findings

Table 1 appends the descriptive item-based results of the appraisal of the self-care agency scale (ASAS), while the scale’s overall mean values are shown in Table 2, once organized in accordance with their academic performance records reported by the study participants. It is observed that the strongest levels of the ASAS are related to higher academic performance reports, even though this constitutes a single performance (non-qualitative) indicator.

### 3.3. Correlations between SOC, ASAS, and Related Factors

Figure 1 shows the graphical (bivariate) trends among age, academic performance, the ASAS, and the SOC scores, depicting the potential existence of consistent bivariate trends between pairs of variables, to be corroborated through Pearson’s r tests in the subsequent paragraph.

The Pearson’s tests show how there is a direct, positive, and statistically significant correlation between the overall ASAS and SOC scores, even though their magnitude does not exceed the *r* = 0.318 obtained for the relationship between the SOC and ASAS scores (see Table 3). Further, the correlations between the sense of coherence, the self-care agency, and the related variables (age and self-reported academic performance) of the participants as a whole were also significant but with low-to-moderate magnitudes.

### 3.4. Differences in Academic Performance according to Individuals’ SOC and the ASAS

Finally, the analysis of variance (One-way ANOVA) showed significant results for the academic performance self-reported by our study partakers when using the SOC and ASAS degrees (below or above the median) as categorical variables useful to group the individuals. The results of the analysis are available in Table 4, showing that the overall academic performance scores tend to be significantly greater among the participants, with both SOC and ASAS scores equal to or above the 50th percentile of the sample distribution.

## 4. Discussion

The core aim of this study was to investigate the potential relationship between the appraisal of the self-care agency and the sense of coherence as mediating variables of mental health and wellbeing. Overall, the obtained results support the assumption of a consistent relationship between these two constructs, adding up to the possibility of differentiate relevant outcomes, such as nursing students’ academic success, on the basis of their ASAS and SOC outcomes.

Coherently with the general findings of this study, the literature suggests that there is a significant and positive relationship between both concepts: the greater the SOC, the greater the ASAS [70,71]. Moreover, the sense of coherence was found to positively impact one’s motivation for self-care [31,72,73]. Such correlation seems to be also corroborated by our results, as we can see in Figure 1 and Table 3. In addition, we have also found that middle-aged women performing higher academic results develop better self-care levels, and these students also obtained more powerful scores in their SOC, as the findings of the major systematic review on the SOC in nurses suggest [74]. More investigations report that those students showing strong SOC values are more resistant to stress, keep healthier behaviours, show higher professional engagement, and, furthermore, theoretical assumptions determine that they do better academically [75,76,77,78].

According to previous evidence, a SOC has been recognized as an influential variable in healthcare-related professionals’ occupational health, finding a strong positive association with greater work engagement and better management of job conflicts [37]. All these data suggest that future university graduate students should be provided with better self-care so that they will be able to cope satisfactorily with the stressors that the nursing career entails. This evidence also suggests that, indeed, they could suffer less incidence of mental health and chronic health conditions if they enjoy better control of self-care guidelines presenting better indicators of stress control [18,79,80].

In addition, some other previous research outcomes dealing with health and welfare-related issues among similar populations suggest that health professionals scoring higher in the SOC acquire an improved self-care agency, resulting in a more comprehensive response to negative life habits’ impacts [81,82]. In addition, they better manage the control patterns in the case of chronic diseases [14,18,83,84]. Likewise, other studies appeal to these types of health professionals and caregivers (CNAs, registered nurses, nurse practitioners, advance practice nurses, etc.) as indispensable for providing education in health patterns and good-care guidelines to their patients and users so that their positioning within the salutogenic model framework and its application in professional practice becomes paramount [85,86,87]. This addresses the need to change the culture and philosophy of care towards salutogenic health systems, paying credence to the nested complexity of human health and striving to strike an adaptive balance between health production and the provision of medical care [30]. Taking into account that nurses are considered reference care agents and health bellwether figures for public health systems, it seems to be necessary to identify these care professionals with the risk of a weak SOC and low self-care agency in order to carry out interventions aimed at improving their health status initially. Consequently, regarding salutogenic designs for health professionals’ training programmes at different levels, it is also vital to focus on the impact they have on transferring care patterns, especially in the case of chronic health conditions and their impact on mental health [31,73,88,89].

One of the very clear observations in our study, and also suggested in other ones [12,77,78,90], is that the salutogenic orientation, including the SOC exploration, is rarely used explicitly in the context of nursing faculties or health vocational and educational training centers. There is a need for research in this field from students’ stages to reinforce acceptable levels of a sense of coherence and proper self-care agency. Improving both constructs will not only somehow favour a profitable academic performance, a reliable predictor of future professional performance, but it will also allow us to generate and keep a better health status, mental health, and wellbeing. Accordingly, Vinje et al. [91] advocate for facilitating the student to develop the capacity to manage herself in the salutogenic way, which is to identify and mobilize resources to maintain and improve one’s health, and this is a form of self-care. However, the most prominent challenge implied in this salutogenic perspective is the development of a professional care model oriented towards self-care, patient empowerment, and the enhancement of the professional-based (curative, preventive, protective, and promotive) practice. When using this framework, health professionals become change agents and role models for the population they serve [59,60,92].

In the spirit of our study’s discussion, preliminarily, it allows us to establish a positive associative pattern between the ASAS and SOC, potentially mediated by age (as a non-modifiable factor). Notwithstanding, this could only be hypothesized, as this mediation was not possible to be analyzed under a statistical approach as a consequence of the lack of measures on further independent variables (e.g., psychological welfare/distress, life satisfaction, psychosocial resources, and lifestyle factors) [39,42,50]. However, what is possible to hypothesize is that this positive pattern could facilitate the design of interventions from the first stages of the curricula by considering the student as an individual subject to supply skills for self-care, explore resources to experience wellbeing and mental health, and leading to the development of subsequent professional confidence; thus, by extension, a future health professional who will be able to provide “better care” to others, as the evidence and literature confirm [52,54,93]. Based on the presented findings, advocating for a salutogenic orientation from a nursing perspective focused on the identification of mental wellbeing factors and fostering self-care must be a current priority in nursing academics. As a final consideration, it is paramount to ensure that teachers are trained in salutogenic approaches for shaping the salutogenic paradigm by means of educative interventions [94] in order to guarantee a successful implementation of these pioneer pedagogical models.

### Limitations and Further Research

According to the validity of the instruments used in this study, the literature has widely shown that the SOC questionnaire and ASAS are reliable and valid instruments to assess both constructs in the Spanish population. The study sample was considerably large, and all the basic statistical parameters were successfully met during the analysis phase. Notwithstanding, there are some potential limitations related to the cross-sectional study design, which hamper the ability to make causal inferences.

First of all, and as our number of core study variables was considerably limited, the procedures such as the linear and multinomial regressions used to predict the academic performance through these factors remained very short in terms of the explained variance, achieving, at best, very small percentages of it (around 8%) in the absence of more related measures useful as predictors. Further, and even when questionnaire-based designs help in terms of practicality and accessibility to large samples of respondents, they may enhance several types of bias, including a common method variance and acquiescent responses that are rather typical of self-reported information on topics that may result in being sensitive for participants. Also, it is worth noting that the magnitudes of the statistical test-based coefficients (especially the bivariate correlations) were considerably moderate, even though they may acquire a significant character thanks to, among other factors, the relatively high sample size, the aforementioned potential bias, and (why not) the actual relationships among the variables measured.

It would be of interest to continue testing the proposed model and expand the study to specific samples with different nurses’ features and professional profiles (CNA, RN, and advanced practice nurses) and in active employment nurses in order to obtain adaptations adjusted to these contexts.

As for further research in the field, the evolution of scoring in the sense of coherence and self-care agency over time, at the beginning and at the end of the professional or undergraduate nursing university grade and assistant nursing (vocational and educational training careers), as well as after several years of professional performance through a time series, would be considered as undeniable researching interests. Another future research line to explore would be the inclusion of content on salutogenesis, the sense of coherence, and the appraisal of self-care agency in the CNAs’ study plans and curricula for the purpose of observing the pedagogical intervention. Efforts should be focused on the implementation of holistic self-care promotion projects, combined with the determination of the SOC and ASAS through the before–after designs with a control group within the population study.

## 5. Conclusions

Overall, the results of this study showed high levels of a sense of coherence and self-care agency among the target population (i.e., the certificated nursing assistant (CNA) Spanish students). Also, it was possible to establish a significant and positive relationship between both constructs. Moreover, significant differences in terms of academic performance were found when using both the SOC and ASAS scores as the splitting factors to group the students, being those individuals with greater SOC and/or ASAS mean scores than those self-reporting better academic outcomes.

Coherently, and supported by the current literature, it would be possible to expect that both constructs might work as possible predictor variables of well-being and health in further studies. In practical terms, and in order to strengthen the coherent and person-centered care models in the academic curricula based on the SMH, it is promising to favour this integrated approach aimed at enhancing the SOC and ASAS individual levels as protectors against stressful events, chronic health conditions, and mental illnesses. A salutogenic model for health and care in nursing education not only enables healthier professional staff, but it might probably contribute to the promotion of more realistic care practices for individuals and communities that should acquire active habits in the responsibility for the maintenance of their health and well-being.

## Figures and Tables

**Figure 1 ijerph-19-09482-f001:**
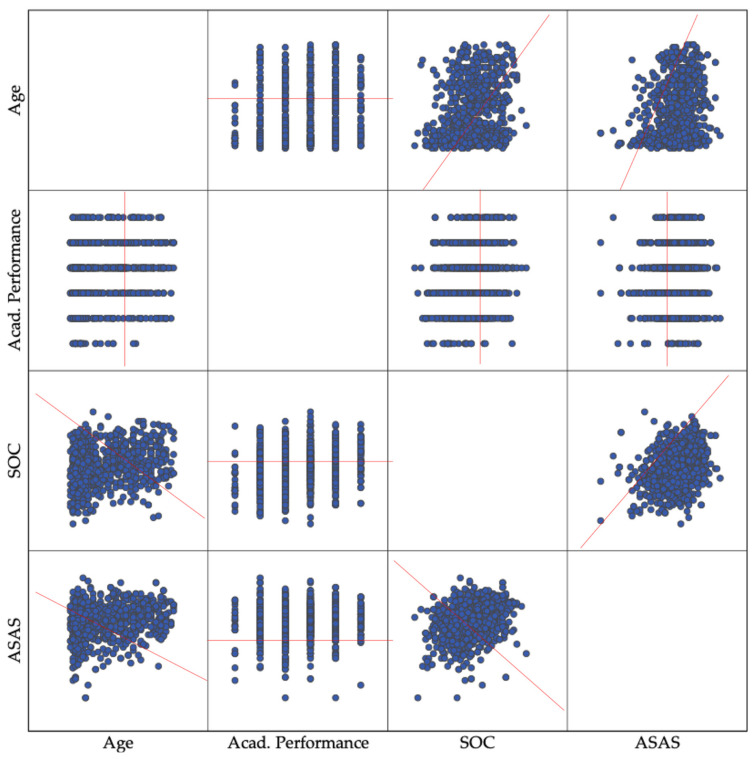
Graphical correlations between the SOC-13 and ASAS total score values per participant, also crossing their scores with participants’ age and self-reported academic performance (n = 921).

**Table 1 ijerph-19-09482-t001:** Item descriptive mean scores of the ASA scale.

Number Item	Item	Mean	SD
ASA1	Make adjustments to stay healthy	4.010	0.923
ASA2	Rarely check measures taken	2.345	1.018
ASA3	Make adjustments to mobility	4.180	1.023
ASA4	Take measures regarding environment	4.430	0.813
ASA5	Set new priorities to stay healthy	3.930	0.952
ASA6	Often lack energy for self-care	3.623	1.326
ASA7	Look for better ways for self-care	3.790	1.149
ASA8	Adjust bathing/showering	4.790	0.649
ASA9	Maintain body weight	3.450	1.274
ASA10	Manage to be myself	4.010	0.950
ASA11	Never include exercise	2.688	1.354
ASA12	Have a circle of friends	4.150	1.144
ASA13	Rarely get enough sleep	2.615	1.249
ASA14	Seldom ask clarification	4.368	0.909
ASA15	Seldom examine body	4.031	1.030
ASA16	Obtain information on medication side effects	3.800	1.124
ASA17	Have changed old habits	3.290	1.380
ASA18	Take safety measures	4.310	0.894
ASA19	Evaluate effectiveness	3.940	0.944
ASA20	Seldom care for myself	2.999	1.273
ASA21	Get information needed	4.250	0.909
ASA22	Seek help	4.040	1.048
ASA23	Seldom have time	2.245	1.128
ASA24	Unable to care for myself	3.884	0.988

**Table 2 ijerph-19-09482-t002:** The ASAS scores and components according to academic performance.

Academic Performance	n	Minimum	Maximum	Mean	SD
0- Fail	25	55	106	88.560	13.520
1- Pass	119	63	120	92.390	10.649
2- Good	242	44	113	89.822	11.200
3- Remarkable	303	56	118	92.630	9.659
4- Outstanding	179	44	114	94.000	10.268
5- With Merit	53	52	108	93.965	9.454

**Table 3 ijerph-19-09482-t003:** Bivariate (Pearson) correlations among study variables.

Variable		1	2	3	4
**1**	Age (years)	--	0.242 **	0.272 **	0.223 **
**2**	Academic Performance		--	0.219 **	0.113 **
**3**	Sense of Coherence (SOC)			--	0.318 **
**4**	Self-care Agency (ASAS)				1

Notes: The correlation is significant at the level ** *p* < 0.001 (bilateral).

**Table 4 ijerph-19-09482-t004:** One-way ANOVA tests for academic performance, grouping participants by the SOC and ASAS degrees.

Variable	Group ^a^	Mean	SD ^b^	SE ^c^	*F* ^d^	*p*
Sense of Coherence (SOC)	Greater SOC	2.48	1.18	0.056	33.439	<0.001
	Lower SOC	2.92	1.12	0.051		
Self-care Agency (ASAS)	Greater ASAS	2.61	1.11	0.053	6.513	0.011
	Lower ASAS	2.80	1.20	0.056		

Notes: ^a^ Group to which each participant belongs (divided by the 50th percentile); ^b^ SD = Standard Deviation; ^c^ SE = Standard Error; ^d^
*F* = *F*-test outcome.

## Data Availability

Not applicable.

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
