# Peer review of "Self-Care and Sense of Coherence: A Salutogenic Model for Health and Care in Nursing Education"

_ijerph, 2022, doi:10.3390/ijerph19159482_

Round 1

Reviewer 1 Report

Line 19 Abstract : Mat removal titles" Background, etc...word usage -of sense of coherence. Perhaps " For sense of Coherence"

26-Remove "methods , just describe your methodology segment same with results...etc. 

29-Flow of word usage 

32- Perhaps word usage pay attention ( remove) try recognize instead .

34-Pronoun "we" is used, remove. 

40-Very informative and descriptive. Interesting concepts and connections

74-Great descriptors / definitions

77-Add the year in citation 

170-First , not firstly 

180-Great details and well written 

210-Perhaps outlining SPSS and definition 

220-Ethics-Ass more detail , data storage , confidentiality etc...Will data be destroyed after number years, where is being stored and by whom ? 

250-Great visuals charting and diagram

302-IMplications training Great 

320-May add more implications and recommendations future study . Build on conclusion section. References

Very interesting study ...LML

Author Response

Response to Editor and Reviewers’ Comments
First of all, thank you for your positive appreciation of our work. Below, you will find our responses to the comments appended in your reviews.
Also, please note that the modifications made in the manuscript will also be tracked (written in green) at the revised paper.
Response to reviewers
Thank you for taking the time to review our manuscript. We are very grateful for the reviewers’ comments on our paper. We have considered them with care, and the comments have been valuable for us when improving the manuscript. Please find in the document the comments, our responses and changes made. 

Reviewer 2 Report

1. This research is important and practical and deserves publication.

2. "Table 3 Correlation of SOC, ASAS and related factors". The correlation data of SOC and other factors didn’t list in Table 3, please add.

3. Gender in Table 3 is a categorical variable and should not be correlated with other variables. Please modify the statistical method.

4. The analytical procedure and results of the Line 257-261 One-way ANOVA didn’t clearly state. Please explain clearly.

Author Response

(The authors gave the same response as above.)

Reviewer 3 Report

thank you for nice research on this important topic; what is missing is more details, for example, you can simply have a focus group from the participants, to provide more details about this self care needs, you have identified, what they are doing to take care of themselves, but not the how, or the important details....

Author Response

(The authors gave the same response as above.)

Reviewer 4 Report

The study used a salutogenic approach to show Sense of Coherence and Self-care Agency Scale in nursing students, which is interesting and innovative, that could increase attention and improve students’ health and wellbeing. Some minor revision needed as below. 

1. Page 1 line 28, abbreviation of appraisal of self-care agency should only be ASA. ASAS should be abbreviation of Appraisal of Self-care Agency Scale. The authors should keep them consistent throughout the manuscript. When discussing correlation, it should always be ASAS, the scale/score, not ASA, the agency. 

2. Page 7 line 251, it’s a new paragraph, better to use “3.3 Correlations between SOC, ASAS and related factors”. In Table 3, no data showed SOC, how they calculated correlations between SOC and others? Why there is correlation between the Age and gender (0.086), what does it mean? 

3. In addition to Table 3, the author may also provide the correlation curves for each one (similar to Figure 1), at least in supplementary results. It hard to tell different between ASAS and Academic performance from Table 2. Therefore, the authors need to provide more evidence to support Table 3, which is the main conclusion of this study. 

Author Response

(The authors gave the same response as above.)

Reviewer 5 Report

First of all, thanks for the opportunity to review this article.

The objective of this study was to analyze the relationship between self-care agency and sense of coherence with age, gender and academic performance. To do this, the authors carried out a cross-sectional study in a sample of 921 Spanish nursing assistants.

In general, the study is well justified, with a good introduction where the important concepts are exposed and related. The methods seem adequate to respond to the objective and the results are widely discussed with the literature. However, I think there are some aspects that authors should consider:

General features:

- Please, whenever possible, authors should try not to cut tables at the end of the page (i.e. Table 1)

- Line 260-261: looks like original template text.

Methods:

- The authors describe the study as analytical. It is true that age and sex are variables that are given, but not academic performance. So, I think the authors should reconsider this description. In fact, the authors themselves establish in the limitations section some aspects for which it is not possible to establish causal relationships.

- The authors include a calculation of the sample size, but the sampling is not probabilistic. This is not consistent. Better to talk about response rates than sample size.

- It is true that the analytical strategy fits the proposed objective, but I think that the authors should move forward with the use of linear regressions, for example.

Results:

- I think it is convenient to improve the explanation of the ANOVA results to make it easier for readers to understand.

Discussion:

The correlations are significant, but do not seem relevant. Only one has a correlation coefficient of 40%. This should be highlighted and discussed.

Conclusions:

The conclusions are not based on the results obtained, but possible relationships with variables that were not studied are inferred from the results of other studies included in the discussion. Please, the authors should give a direct answer to the objective of the study.

I hope these comments are helpful.

Best regards.

Author Response

(The authors gave the same response as above.)

Round 2

Reviewer 5 Report

First, I thank you again for the opportunity to review this work.

I believe that the authors did an excellent job revising their article and I consider that this paper meets the necessary requirements to be published.

Congratulations!

My best wishes.